

Effects of the dynamic effective porosity on watertable fluctuations and seawater
intrusion in coastal unconfined aquifers
Zhaoyang Luo[1,2,#], Jun Kong[1,3,#], Lili Yao[4], Chunhui Lu[1], Ling Li[5,6], David Andrew Barry[2]
[1]State Key Laboratory of Hydrology-Water Resources and Hydraulic Engineering, Hohai
University, Nanjing, China
[2]Ecological Engineering Laboratory (ECOL), Environmental Engineering Institute (IIE),
Faculty of Architecture, Civil and Environmental Engineering (ENAC), École Polytechnique
Fédérale de Lausanne (EPFL), Lausanne, Switzerland
[3]Key Laboratory of Coastal Disaster and Protection (Ministry of Education), Hohai
University, Nanjing, China
[4]Department of Civil, Environmental, and Construction Engineering, University of Central
Florida, Orlando, USA
[5]School of Engineering, Westlake University, Hangzhou, China
[6]School of Civil Engineering, The University of Queensland, Brisbane, Australia
[#]Corresponding authors: Zhaoyang Luo (zhaoyang.luo@epfl.ch) and Jun Kong
(kongjun999@126.com)





**Abstract**

Watertable fluctuations and seawater intrusion are characteristic features of coastal

unconfined aquifers. The dynamic effective porosity due to watertable fluctuations is analyzed

and then a modified (empirical) expression is proposed for the dynamic effective porosity

based on a dimensionless parameter related to the watertable fluctuation frequency. After

validation with both experimental data and numerical simulations, the new expression is

implemented in existing Boussinesq equations and a numerical model, allowing for

examination of the effects of the dynamic effective porosity on watertable fluctuations and

seawater intrusion in coastal unconfined aquifers, respectively. Results show that the

Boussinesq equation accounting for the vertical flow in the saturated zone and dynamic

effective porosity can accurately predict experimental dispersion relations (that all existing

theories fail to predict), highlighting the importance of the dynamic effective porosity in

modeling watertable fluctuations in coastal unconfined aquifers. This in turn confirms the

utility of the real-valued expression of the dynamic effective porosity. An outcome is that the

phase lag between the total moisture (above the watertable) and watertable height measured in

laboratory experiments using vertical soil columns (1D systems) can be ignored when

predicting watertable fluctuations in coastal unconfined aquifers (2D systems). A dynamic

effective porosity that is, by comparison, smaller than the soil porosity leads to a reduction in

vertical water exchange between the saturated and vadose zones and hence watertable waves

can propagate further landward. The dynamic effective porosity further plays a critical role in

simulations of seawater intrusion, since it predicts a more landward seawater-freshwater





interface and a higher position of the upper saline plume.
**Keywords:** Dynamic effective porosity; Boussinesq equation; watertable fluctuation;
dispersion relation; seawater intrusion


**Highlights**

➢ A modified expression is proposed for the dynamic effective porosity due to watertable
fluctuations
➢ The Boussinesq equation correctly predicts watertable fluctuations with involving the
vertical flow and dynamic effective porosity
➢ Numerical models further underestimate seawater intrusion if ignoring the dynamic
effective porosity



## 1. Introduction

As a transition zone between the ocean and land, coastal unconfined aquifers respond to interactions between terrestrial fresh groundwater and seawater. Due to oceanic oscillations (e.g., tides or waves), water flows into or out of the aquifer periodically, which directly affects a range of groundwater-dependent processes including sediment mobilization, seawater intrusion, submarine groundwater discharge, solute transport and chemical loading to the ocean (e.g., Parlange et al., 1984; Li et al., 1999; Moore, 2010; Xin et al., 2010; Bakhtyar et al., 2011; Werner et al., 2013; Robinson et al., 2018; Wallace et al., 2020). Watertable fluctuations induced by oceanic oscillations are an important signal for quantifying these processes, and their prediction is a longstanding topic (e.g., Philip, 1973; Smiles & Stokes, 1976; Parlange et al., 1984). They are investigated by field measurements (e.g., Nielsen et al., 1990; Raubenheimer et al., 1999; Robinson et al., 2006; Heiss & Michael, 2014; Trglavcnik et al., 2018), laboratory experiments (e.g., Cartwright et al., 2004; Robinson & Li, 2004; Shoushtari et al., 2016, 2017), numerical simulations (e.g., Li et al., 1997; Cartwright et al., 2006; Shoushtari et al., 2015; Brakenhoff et al., 2019) and analytical solutions (e.g., Parlange & Brutsaert, 1987; Barry et al., 1996; Nielsen et al., 1997; Li et al., 2000a,b; Teo et al., 2003; Song et al., 2007; Kong et al., 2013, 2015). Among these, analytical solutions based on the 1D Boussinesq equation describe watertable fluctuations and give results that are easily computed, present explicit relations between parameters that impact watertable fluctuations, and can be used as benchmarks for numerical simulations.

The Dupuit-Forchheimer based Boussinesq equation describes groundwater flow in the



saturated zone (Bear, 2012). It is amenable to analytical investigations, and so is used to
reveal mechanisms that influence watertable wave propagation (e.g., Nielsen, 1990; Li et al.,
2000a; Teo et al., 2003; Jeng et al., 2005a). To underline the effects of vertical flow on
watertable fluctuations, Nielsen et al. (1997) and Liu and Wen (1997) proposed an
intermediate-depth Boussinesq equation based on different approximation methods. However,
all these investigations ignore unsaturated flow that arises when the watertable rises and falls,
which affects predictions of watertable fluctuations (e.g., Gillham, 1984; Kong et al., 2016;
Luo et al., 2018). To improve such predictions, modifications to the Boussinesq equation are
used. An early step in this direction was taken by Parlange and Brutsaert (1987), who
presented a Boussinesq equation that included vertical exchange between the saturated and
vadose zones. Barry et al. (1996) presented an approximate analytical solution that considered
the effect of this exchange on watertable fluctuations for the case of a periodic boundary
condition. Following the method of Parlange and Brutsaert (1987), Jeng et al. (2005b) further
improved the Boussinesq model by proposing a higher-order capillarity correction. These
modifications were all based on the 1D Boussinesq equation (horizontal flow only) without
explicit consideration of the vertical flow (although the effect of the unsaturated zone is
considered indirectly). The combined effects of unsaturated and vertical flows were
investigated by Li et al. (2000b) and Shoushtari et al. (2016), who extended the intermediate-
depth Boussinesq model proposed by Nielsen et al. (1997). Recently, Kong et al. (2013, 2015)
developed two types of Boussinesq equation involving horizontal unsaturated flow under the
assumption of a hydrostatic vertical pressure distribution, accounting for both short (Kong et



al., 2013) and long (Kong et al., 2015) period watertable fluctuations.
When applied to analyze watertable fluctuations in coastal unconfined aquifers subjected
to archetypal case of a single-component boundary fluctuation, most of the abovementioned
Boussinesq equations predict an asymptotic amplitude decay rate and zero phase lag increase
rate (standing wave behavior) of watertable waves for increasing dimensionless aquifer depth,
$n_e\omega D/K_s$ (where $n_e$ [-] is the static effective porosity, $\omega$ [T$^{-1}$] is the angular frequency of the
boundary forcing, $D$ [L] is the aquifer depth and $K_s$ [LT$^{-1}$] is the saturated hydraulic
conductivity) (e.g., Barry et al., 1996; Liu & Wen, 1997; Nielsen et al., 1997; Li et al., 2000a;
Kong et al., 2013, 2015). Nevertheless, the laboratory experiments of Shoushtari et al. (2016),
which cover a wide range of $n_e\omega D/K_s$ in an unconfined aquifer with a vertical boundary,
indicated an increase of both the amplitude decay rate and phase lag increase rate of
watertable waves with increasing $n_e\omega D/K_s$. They concluded that all existing Boussinesq
equations cannot predict experimental results correctly. This discrepancy between
experimental data and theoretical predictions, especially for the phase lag increase rate,
motivates further investigations on the potential mechanisms controlling watertable
fluctuations in coastal unconfined aquifers.
Few of the abovementioned Boussinesq equations consider the dynamic effective
porosity observed during watertable fluctuations (Nielsen & Perrochet, 2000; Cartwright et
al., 2005; Acharya et al., 2012; Pozdniakov et al., 2019). Shoushtari et al. (2016) combined
the complex, frequency-dependent effective porosity proposed by Cartwright et al. (2005)
with the infinite-order Boussinesq equation of Nielsen et al. (1997), but their new theory



failed to predict the experimental results. Therefore, they did not further analyze the effects of

the dynamic effective porosity on watertable fluctuations in detail, and it remains unclear

whether the dynamic effective porosity affects watertable fluctuations. In addition, as

indicated by Hilberts and Troch (2006), the complex-valued expression for the dynamic

effective porosity proposed by Cartwright et al. (2005) has limited practical use.

Watertable fluctuations are of additional interest since they affect seawater intrusion

behavior in coastal unconfined aquifers. In such aquifers, density-driven flow leads to two

different salt areas: a saltwater wedge and an upper saline plume (Robinson et al., 2007).

Seawater intrusion under watertable fluctuations is usually investigated numerically (e.g.,

Brovelli et al., 2007; Xin et al., 2010; Liu et al., 2014; Robinson et al., 2014; Levanon et al.,

2016; Yu et al., 2019). Levanon et al. (2017) investigated the responses of seawater intrusion

and watertable to surface water level fluctuations combining field measurements with

numerical simulations. More recently, Fang et al. (2021) examined the response of seawater

intrusion to tide-induced unstable flow using laboratory experiments and numerical

simulations. For more information regarding this topic, readers are referred to the

comprehensive reviews of Robinson et al. (2018) and Werner et al. (2013). Models of density-

dependent flow in coastal aquifers, like the abovementioned Boussinesq models, treat the

effective porosity as a constant.

Here, the effects of the dynamic effective porosity effects on watertable fluctuations and

seawater intrusion are further explored. First, it is determined empirically from existing

experimental measurements and numerical results. Then, modified governing equations that



consider the effects of the dynamic effective porosity are proposed to predict watertable
fluctuations. This allows for exploration of the underlying mechanisms causing the
discrepancy between experimental data and existing theoretical predictions for watertable
fluctuations. Since saltwater intrusion is affected by watertable fluctuations, we further
examine to what extent the dynamic effective porosity will affect saltwater intrusion in coastal
unconfined aquifers based on numerical simulations.

## 2. Methods

### 2.1 Original Governing Equations for Watertable Fluctuations

Under the assumption that streamlines are parallel to the underlying bedrock (assumed
horizontal) and neglecting unsaturated flow, watertable fluctuations (due, e.g., to tidal forcing)
in a coastal unconfined aquifer (supporting information Figure S1) can be described by,

$$n_e \frac{\partial h}{\partial t} = K_s \frac{\partial}{\partial x}\left( h \frac{\partial h}{\partial x} \right) \tag{1}$$

where $t$ [T] is time, $x$ [L] is the horizontal distance from the vertical seaward boundary, and $h$
[L] is the watertable elevation above the aquifer base. Bear (2012) discusses the physical
basis of equation (1), often called the Boussinesq equation in groundwater literature (e.g.,
Hogarth & Parlange, 1999; Parlange et al., 2001). To account for the effects of vertical flow
during watertable fluctuations, equation (1) is modified as (Liu & Wen, 1997),

$$n_e \frac{\partial h}{\partial t} = K_s \frac{\partial}{\partial x}\left( h \frac{\partial h}{\partial x} \right) + \frac{K_s D^3}{3} \frac{\partial^4 h}{\partial x^4} \tag{2}$$

Setting $\eta = h - D$ and assuming $\eta \ll D$, equations (1) and (2) are respectively linearized to
(Liu & Wen, 1997),





$$n_e \frac{\partial \eta}{\partial t} = K_s D \frac{\partial^2 \eta}{\partial x^2}$$
(3)

$$n_e \frac{\partial \eta}{\partial t} = K_s D \left( \frac{\partial^2 \eta}{\partial x^2} + \frac{D^2}{3} \frac{\partial^4 \eta}{\partial x^4} \right)$$
(4)

Note that equation (4) is the same as the linearized form of the second-order approximation of
Nielsen et al. (1997). For the archetypal case of a single-component sinusoidal tide at the sea
boundary (semi-infinite domain, perpendicular beach at the sea boundary, Figure S1), the
corresponding dispersion relations of watertable waves for equations (3) and (4) are,
respectively (Liu & Wen, 1997; Nielsen et al., 1997; Shoushtari et al., 2016),
$$kD = \sqrt{\frac{n_e \omega D}{K_s} i}$$
(5)

$$kD = \sqrt{\frac{3}{2}} \sqrt{-1 + \sqrt{1 + \frac{4}{3} \frac{n_e \omega D}{K_s} i}}$$
(6)

where $k = k_r + ik_i$ is the watertable wave number with $i = \sqrt{-1}$. The real ($k_rD$) and imaginary
($k_iD$) parts represent the amplitude spatial decay rate and phase lag increase rate of watertable
waves, respectively (Liu & Wen, 1997).
**2.2 Relation between the Effective Porosity and Fluctuation Frequency**

The effective porosity is defined as the volume of water that an unconfined aquifer

releases or gains per unit surface area of aquifer per unit change of the watertable height
(Childs, 1960). In most existing Boussinesq equations, the effective porosity is treated as a
soil-dependent constant (e.g., Barry et al., 1996; Nielsen et al., 1997; Kong et al., 2013, 2015).
However, experimental (Nielsen & Perrochet, 2000; Cartwright et al., 2005), numerical
(Acharya et al., 2012; Pazdniakov et al., 2019) and field (Rabinovich et al., 2015) evidence





indicates that the effective porosity is dynamic and may depend on the porewater velocity.
Recently, Pazdniakov et al. (2019) proposed an approximate expression to predict the
dynamic effective porosity under seasonal or diurnal groundwater level fluctuations.
Motivated by the results of Cartwright et al. (2005) and Pazdniakov et al. (2019) and
assuming no truncation of the capillary fringe, we introduce a modified empirical expression
to describe the relation between the dynamic effective porosity and fluctuation frequency,

$$n_t = n_e \left[ 1 - \exp\left[ -\left( \frac{a}{\tau_\omega} \right)^b \right] \right] \tag{7a}$$

with

$$\tau_\omega = \frac{n_e H_\psi / K_s}{1 / \omega} = \frac{n_e \omega H_\psi}{K_s} \tag{7b}$$

$$n_e H_\psi = \int_0^\infty (\theta - \theta_r) d\psi \tag{7c}$$

where $n_t$ [-] is the dynamic effective porosity, $\tau_\omega$ [-] is a dimensionless parameter related to
the watertable fluctuation frequency and soil properties, $a$ [-] and $b$ [-] are the fitting
parameters (Section 3.1), $\theta$ [-] is the soil water content related to the capillary suction $\psi$
[L], $\theta_r$ [-] is the residual soil water content, and $H_\psi$ [L] is a measure of the equivalent
saturated height of the unsaturated zone (Parlange & Brutsaert, 1987; Cartwright et al., 2005).
Note that equation (7a) has the same functional form as that of Pazdniakov et al. (2019) but a
with different variable $\tau_\omega$. We will discuss equation (7a) in more detail below.

To solve equation (7c), the relation between $\theta$ and $\psi$ is described by a modified van

Genuchten model (Troch, 1993; Kong et al., 2016; Luo et al., 2019),

$$\theta = (\theta_s - \theta_r) S_e + \theta_r = (\theta_s - \theta_r) \left[ 1 + (\alpha_1 \psi)^{n_1} \right]^{-1 - 1/n_1} + \theta_r \tag{8}$$





where $\theta_s$ [-] is the saturated soil water content, $S_e$ is the effective saturation, and $\alpha_1$ [L$^{-1}$]
and $n_1$ [-] are the parameters obtained by fitting equation (8) to measurements of the soil
moisture characteristic curve. Other expressions (e.g., van Genuchten, 1980) can also be used
to describe the relation between $\theta$ and $\psi$, but equation (7c) may then need to be evaluated
numerically. Observe that the difference between equation (8) and the original van Genuchten
(1980) model (VG model) is the exponent, which makes equation (8) integrable to attain
simple analytical expression, i.e., substituting equation (8) into equation (7c) leads to,

$$H_\psi = \frac{1}{\alpha_1} \tag{9}$$


By comparison, Pazdniakov et al. (2019) derived $\tau_\omega$ from the relation between the
hydraulic conductivity and the capillary suction, whereas here it is derived from the relation
between soil water content and capillary suction. In addition, $\tau_\omega$ is an approximation in
Pozdniakov et al. (2019) (their equation (12)), whereas here it is an exact expression (due to
use of equation (8)) related to the fluctuation period and porewater velocity.

**2.3 Modified Governing Equations that Consider the Dynamic Effective Porosity**


By replacing $n_e$ in equations (3) and (4) with $n_t$ from equation (7a), the governing
equations for watertable fluctuations become, respectively,

$$n_e\left[1 - \exp\left[-\left(\frac{a}{\tau_\omega}\right)^b\right]\right]\frac{\partial \eta}{\partial t} = K_s D \frac{\partial^2 \eta}{\partial x^2} \tag{10}$$


$$n_e\left[1 - \exp\left[-\left(\frac{a}{\tau_\omega}\right)^b\right]\right]\frac{\partial \eta}{\partial t} = K_s D \left(\frac{\partial^2 \eta}{\partial x^2} + \frac{D^2}{3}\frac{\partial^4 \eta}{\partial x^4}\right) \tag{11}$$






The corresponding dispersion relations of watertable waves from equations (10) and (11)
are, respectively,

$$kD = \sqrt{\left[1-\exp\left[-\left(\frac{a}{\tau_\omega}\right)^b\right]\right]\frac{n_e\omega D}{K_s}i} \tag{12}$$

$$kD = \sqrt{\frac{3}{2}}\sqrt{-1+\sqrt{1+\frac{4}{3}\left[1-\exp\left[-\left(\frac{a}{\tau_\omega}\right)^b\right]\right]\frac{n_e\omega D}{K_s}i}} \tag{13}$$

Below, the effects of the dynamic effective porosity on watertable fluctuations will be
examined. Consistent with previous studies, possible truncation of the capillary fringe by the
soil surface is ignored when examining watertable fluctuations using the Boussinesq equation
(e.g., Parlange & Brutsaert, 1987; Barry et al., 1996; Li et al., 2000; Song et al., 2007; Kong
et al., 2015).

## 3. Results and Discussion

### 3.1. Effects of Watertable Fluctuations on the Dynamic Effective Porosity: 1D Column Experiments

The main reason that the dynamic effective porosity varies with watertable fluctuation
rate is the filling/drainage of the unsaturated zone above the aquifer, which is rate-limited due
to the water flow rate (Li et al., 1997; Cartwright, 2014). Previous experiments to quantify
this exchange used 1D column experiments (Figure S2). For simplicity, readers are referred to
Cartwright et al. (2005) for details about the 1D column experiment. To establish the relation
between the dynamic effective porosity ($n_t$) and forcing fluctuation frequency ($\omega$), the
dimensionless parameter $\tau_\omega$ (equation 7b) is used. Physically, it is the ratio of the response





timescale of the unsaturated zone ($n_e H_\psi / K_s$, the minimum time needed by the unsaturated zone
to fully respond to a boundary disturbance) to the timescale of the surface water level
fluctuation ($1/\omega$). If the response time of the unsaturated zone is much
longer than the timescale of the surface water level fluctuation (i.e., large $\tau_\omega$), the water in
the unsaturated zone will not have sufficient time to reach equilibrium during watertable
fluctuations. Then, the water exchange between the saturated and unsaturated zones will
reduce, thus leading to a smaller $n_t$. This behavior is consistent with experimental evidence
(Cartwright et al., 2005; Cartwright, 2014; Luo et al., 2020) and the complex (dynamic)
effective porosity concept of Nielsen and Perrochet. (2000) and Cartwright et al. (2005).
Equation (7a) captures this behavior since $n_t$ approaches $n_e$ for small $\tau_\omega$, whereas it tends to
zero for large $\tau_\omega$.
The appropriateness of the functional form of equation (7a) is checked by fitting it to $n_t$
obtained from experiments and numerical simulations. Using the similar experimental
apparatus presented in Figure S2, Cartwright et al. (2005) carried out 63 experiments
involving three different soils. For simplicity, readers are referred to Cartwright et al. (2005)
for more details about the experiments. As can be seen from Figure 1a, the range of $n_t/n_e$ and
$\tau_\omega$ for these experiments varies from $10^{-2.5}$ to $10^{-0.4}$ (two orders of magnitude) and $10^{-2}$ to $10^3$
(five orders of magnitude), respectively. As expected, $n_t/n_e$ declines with increasing $\tau_\omega$. In
addition, regardless of soil type there is a clear relation between $n_t/n_e$ and $\tau_\omega$, which can be
fitted (MATLAB© ver. 9 function "lsqcurvefit") by equation (7a) with $a = 0.0335$ and $b =$
0.4444 (Figure 1a). In general, equation (7a) performs well although there is small deviation



between equation (7a) and experimental data at larger $\tau_\omega$. The fitted values match well with
the experimental values ($R^2 = 0.94$), indicating that equation (7a) satisfactorily describes the
dynamic effective porosity as it varies with the fluctuation frequency for different soils.

Since the ranges of $n_t/n_e$ and $\tau_\omega$ obtained from experiments are relatively narrow, we

consider also the numerical simulations of Pozdniakov et al. (2019). A total of 299 cases
covering 100 random soils was simulated with HYDRUS 1D (Šimůnek et al., 2020).
Compared with available experiments, these numerical simulations cover wider ranges of
$n_t/n_e$ and $\tau_\omega$, with the former varying from $10^{-3}$ to $10^0$ (three orders of magnitude) and the
latter from $10^{-3}$ to $10^5$ (eight orders of magnitude). As shown in Figure 2a, $n_t/n_e$ is equal to
unity at small $\tau_\omega$ and then decreases rapidly with increasing $\tau_\omega$, suggesting a frequency-
dependent $n_t/n_e$. Again, numerical results show a clear dependence of $n_t/n_e$ on $\tau_\omega$ although
both $n_t/n_e$ and $\tau_\omega$ span a broad range. For the numerical model results, the relation between
$n_t/n_e$ and $\tau_\omega$ can be approximated by equation (7a) with a small deviation at large $\tau_\omega$ taking
$a = 0.1216$ and $b = 0.3642$. We also plotted the fitted $n_t/n_e$ versus numerical $n_t/n_e$ (Figure 2b).
The fitted values match well with the numerical values ($R^2 = 0.98$).

Based on the experimental and numerical data, the effective porosity will be significantly

impacted by watertable fluctuations when $\tau_\omega$ is larger than 0.01 (green dashed line in
Figures 1a and 2a). This gives a critical value to identify whether the unsaturated zone has
sufficient time to respond to a boundary fluctuation. It is worth noting that the values of $a$ and
$b$ are different for experimental and numerical results, which leads to a small deviation
between the optimal fitting curves (Figure 2a). This systematic deviation could be induced by





either the experiments or the simulations. On the one hand, hysteresis is ignored in the

numerical simulations. On the other hand, there is a lack of measurements for $n_t/n_e$ close to 1

for experiments, which will affect the fitting results. In addition, a linear response of the

watertable is assumed to determine the experimental dynamic effective porosity (Cartwright

et al., 2005), i.e., the governing equation was linearized above under the assumption of a

small amplitude fluctuation, which may contribute to the deviation. Overall, despite these

uncertainties, the relation between the dynamic effective porosity and fluctuation frequency

can be predicted by equation (7a). In practice, we suggest that the fit of equation (7a) to the

experimental data is preferred.

Both the 1D sand column experiments carried out by Nielsen and Perrochet (2000) and

Cartwright et al. (2005) indicate a phase lag between the total moisture (above the watertable)

and watertable height during watertable fluctuations. In order to account for this phase lag,

Cartwright et al. (2005) suggested a complex-valued expression, where the real and imaginary

parts respectively represent the effective porosity and phase lag, to describe the dynamic

effective porosity. However, in this study, we focus on watertable fluctuations in coastal

unconfined aquifers (2D systems) as they respond boundary forcing with a fixed frequency,

not the moisture content above the watertable. For this case, in the following section we

examine the impact of equation (7a) in predicting watertable fluctuations in careful 2D

experiments for horizontal transmission of watertable fluctuations forced by a single-

component watertable change at the boundary.



**3.2. Effects of the Dynamic Effective Porosity on Watertable Fluctuations: 2D Aquifer**

**Experiments**

Here, we use an extensive set of existing 2D experimental results to examine the effects

of the dynamic effective porosity on watertable fluctuations. Specifically, the combination of

equation (7a) with existing Boussinesq equations produced modified governing equations,

equations (10) and (11), which can be used to predict the effects of the dynamic effective

porosity on watertable wave propagation in coastal unconfined aquifers. Following previous

studies (e.g., Nielsen, 1990; Barry et al., 1996; Li et al., 2000a; Cartwright et al., 2004; Kong

et al., 2015), the dispersion relation linking the amplitude decay rate ($k_rD$) with phase lag

increase rate ($k_iD$), is adopted to characterize the propagation of watertable waves. A total of

122 sand flume experiments, covering a wide range of $n_e\omega D/K_s$ values, were conducted by

Shoushtari et al. (2016) and are used here to examine the validity of these predictions. The

experiments were carried out in a sand flume with dimensions of 9 m (length) × 0.15 m

(width) × 1.5 m (height). Readers are referred to Shoushtari et al. (2016) for details about the

sand flume experiment. Since the properties are similar, the two sands adopted in Shoushtari

et al. (2016)'s experiments are considered to have the same VG fitting parameters (van

Genuchten, 1980). As can be seen from Figure 3a, equation (8) matches well with the VG

model based on the parameters listed in Table 1.

Figure 4 of Shoushtari et al. (2016) shows the predictions of the dispersion relations in

equations (5) and (6), which were derived from equations (3) and (4), respectively.

Specifically, their Figure 4 plots predicted $k_rD$ vs $k_iD$ curves and experimental data. Equation





(5) performs poorly whereas equation (6) compares well with the data. Note that plots of $k_rD$
vs $k_iD$ are independent of any specific form selected for $n_e$ since such plots depend on a single
dimensionless parameter, $n_e\omega D/K_s$ (Nielsen et al., 1997; Shoushtari et al., 2016). The role of
$n_e\omega D/K_s$ is instead explored by examining each of $k_rD$ and $k_iD$ as a function of $n_e\omega D/K_s$, as
done in Figure 4, to check the applicability of the modified equations (10) and (11) (i.e.,
approximate analytical solutions of the dispersion relation shown in equations (13) and (14),
respectively). Since there is a small deviation between the best-fit curves for the experimental
and numerical data of $n_t/n_e$ and $\tau_\omega$, two pairs of curves for the two pairs of best-fit $a$ and $b$
values are compared: one pair of values obtained from fitting to experimental data and
another pair obtained from fitting to numerical data. In contrast to the relation between $k_rD$
and $k_iD$ (which, as mentioned, is independent of the functional form of $n_e$), there are large
deviations between predictions of $k_rD$ and $k_iD$ as they vary with $n_e\omega D/K_s$ from governing
equations with or without considering the dynamic effective porosity effects, depending on
the values of $a$ and $b$. Moreover, these deviations increase for increasing $n_e\omega D/K_s$. Again, this
emphasizes that the effective porosity is increasingly influenced by watertable fluctuations as
$n_e\omega D/K_s$ increases. Compared to the original equations (3) and (4) (i.e., approximate
analytical solutions of the dispersion relation shown in equations (5) and (6), respectively)
that assume a constant effective porosity, for a given $n_e\omega D/K_s$, both $k_rD$ and $k_iD$ are smaller
when the dynamic effective porosity is considered. This is because a smaller effective
porosity corresponds to reduced vertical water exchange and hence watertable waves can
propagate further landward, i.e., smaller $k_rD$ and $k_iD$ (Li et al., 2000b).



Shoushtari et al. (2016) analyzed their experimental results and demonstrated that all
existing theories are unable to predict the measured dispersion relation correctly (Figure S3 in
supporting information and Figure 5 in Shoushtari et al. (2016)), even considering different
factors (e.g., capillary effect, hysteresis, porous media deformation (Shoushtari & Cartwright,
2017)). In contrast, equation (11) predicts well the relations between $k_rD$ or $k_iD$ and $n_e\omega D/K_s$,
despite the fact that experimental and numerical results do not agree (and give rise to different
values of $a$ and $b$). The success of equation (11) highlights the significant role played by the
dynamic effective porosity on watertable fluctuations, and so confirms that equation (7a)
fitted by nonlinear least squares can be used to describe the relation between the dynamic
effective porosity and fluctuation frequency. Moreover, this in turn suggests that the phase lag
between the total moisture and watertable height measured in laboratory experiments using
vertical soil columns can be ignored when predicting watertable fluctuations. It should be
noted that substituting the complex-valued expression for the dynamic effective porosity
proposed by Cartwright et al. (2005) into equation (2) cannot predict the measured dispersion
relation well (Figure S4). Furthermore, experiments conducted by Parlange et al. (1984) and
Cartwright et al. (2003) were used to check the validity of equation (11). As can be seen from
Figures 5 and 6, predictions of equation (11) agree well with the measured watertable in both
experiments. The analytical solution of $h$ is given in Appendix based on the dispersion
relation.
By comparison, equation (11) with $a$ and $b$ from experiments ($a = 0.0335$ and $b =$
0.4444) performs better than the corresponding simulation-derived values in predicting



experimental results of Shoushtari et al. (2016) and Cartwright et al. (2003). Therefore, the
experimentally-determined values of $a$ and $b$ are recommended for practical use when
predicting watertable fluctuations in coastal unconfined aquifers.
**3.3. Effects of the Dynamic Effective Porosity on Seawater Intrusion**
As seen above, the dynamic effective porosity and watertable fluctuations are
functionally related, especially for high frequency fluctuations. Therefore, one would
intuitively anticipate that the dynamic effective porosity will also influence seawater intrusion
in coastal unconfined aquifers. To explore the influence of the effects of the dynamic effective
porosity on seawater intrusion, we use SUTRA, a 3D variable-saturation and variable-density
groundwater model (Voss & Provost, 2008).
Numerical simulations were carried out for the experiment of Shen et al. (2020), who
investigated seawater intrusion under watertable fluctuation influences in a sand flume with
dimensions of 7.7 m (length) × 1.2 m (height) × 0.16 m (width). Figure S5 illustrates the
numerical model geometry for the base case, as well as relevant numerical settings. Again,
equation (8) matches perfectly the VG model of Shen et al. (2020) (Figure 3b, parameters
listed in Table 1). A high-frequency water level fluctuation with a period of 240 s was
imposed at the sea boundary, leading to dynamic effective porosities of 0.12 and 0.2
computed using the pairs of $a$ and $b$ obtained above by fitting equation (7a) to experimental
and numerical data, respectively (the static effective porosity is 0.4, Table 1). Only the
effective porosity is changed while other parameters are fixed for these two cases. All
numerical simulations were run for 26 h, with a time step of 4 s. The model domains were



discretized with node spacings of 0.02 and 0.1 m in the horizontal and vertical directions,
respectively, to satisfy the Péclet number stability criterion (Voss & Souza, 1987). Different
mesh schemes were tested to ensure mesh-independent numerical results. The transient
locations of the saltwater wedge and upper saline plume were observed to ensure quasi-steady
state results, defined as when the locations of the saltwater wedge and upper saline plume
remain unchanged after one period. The direct solver in SUTRA (DIRECT) was adopted in
the numerical simulations with convergence tolerances of $10^{-10}$ kg/m/s$^2$ and $10^{-10}$ for solver
iterations during pressure and transport solutions, respectively.

Figure 7 compares the experimental and numerical results, which show two different salt

areas under tidal forcing: a saltwater wedge and an upper saline plume. For the base case with
$n_t = 0.4$ (without considering the dynamic effective porosity, Figure 7b), the numerical model
significantly underestimates the seawater-freshwater interface location while it overestimates
the area of upper saline plume. Once the dynamic effective porosity is considered, however,
the numerical model performs much better in predicting seawater intrusion. For the case with
$n_t = 0.2$ ($a$ and $b$ obtained by fitting equation (7a) to numerical data, Figure 7c), the numerical
model predicts almost the same magnitude of the upper saline plume as the experiment,
despite a small deviation for the seawater-freshwater interface location. In contrast, the
numerical model with $n_t = 0.12$ ($a$ and $b$ obtained by fitting equation (7a) to experimental
data, Figure 7d) predicts more accurately the seawater-freshwater interface location, while it
slightly underestimates the area of the area of upper saline plume. These results highlight the
importance of including the dynamic effective porosity in numerical models for assessing



saltwater intrusion. Since the location of the saltwater wedge is more important for coastal
groundwater management, the experimentally-determined values of *a* and *b* are also
recommended for practical use when predicting seawater intrusion in coastal unconfined
aquifers.

Observe that a smaller effective porosity leads to a more landward seawater-freshwater

interface with a higher position of the upper saline plume (Figure 7a). In the governing flow
equation used in the numerical solution, a decrease of the effective porosity is equivalent to an
increase of the hydraulic conductivity. As a result, the seawater-freshwater interface moves
more landward while the upper saline plume moves toward a higher position. Note that a
landward movement of the bottom saltwater wedge will push the upper saline plume upward
to maintain the outward flow of inland freshwater. Regardless of the values of *a* and *b*
obtained from fitting equation (7a) to measured or numerical data, the numerical results match
well with the experiments (Figure 7a). This again confirms that equation (7a) fitted by
nonlinear least squares can be used to describe the relation between the dynamic effective
porosity and fluctuation frequency. Again, this suggests that the phase lag between the total
moisture and watertable height measured in laboratory experiments using vertical soil
columns can be ignored when predicting seawater intrusion.

In reality, wave-dominated unconfined aquifers are widely distributed along the coast

(e.g., Xin et al., 2010; Robinson et al., 2014). Additionally, high-frequency fluctuations are
usually adopted at the sea boundary when conducting seawater intrusion experiments (e.g.,
Kuan et al., 2012; Yu et al., 2019; Shen et al., 2020). Under these conditions, neglecting the



dynamic effective porosity will lead to an inappropriate estimation of seawater intrusion
based on numerical models. Zhang et al. (2016) found that only an increase of $K_s$ (from 17.28
to 132.3 m/d, one order of magnitude larger) enabled their model to perform well in
predicting groundwater flow and seawater intrusion in a field aquifer. Our results suggest that,
in part at least, this increase in $K_s$ is due to their neglect of the dynamic effective porosity
since, as mentioned earlier, an increase of $K_s$ is equivalent to a decrease of $n_t$ in numerical
models.

**4. Conclusions**

This study evaluates the effects of the dynamic effective porosity on watertable

fluctuations and seawater intrusion in coastal unconfined aquifers. Following Pazdniakov et
al. (2019), we propose a new modified expression to predict the dynamic effective porosity
under watertable fluctuations. After validating against both experiments and numerical
simulations, this expression is coupled with existing governing equations and a numerical
model to examine the effects of the dynamic effective porosity on watertable fluctuations and
seawater intrusion, respectively. The results lead to the following conclusions:

(1) The new modified expression is able to predict the dynamic effective porosity

accurately during watertable fluctuations. Moreover, coupling this real-valued expression of
the dynamic effective porosity with existing governing equations leads to accurate predictions
of watertable fluctuations, suggesting the phase lag between the total moisture (above the
watertable) and watertable height measured in laboratory experiments using vertical soil
columns (1D systems) can be ignored when predicting watertable fluctuations in coastal





unconfined aquifers (2D systems). In other words, this work demonstrates that a real-valued
expression for the dynamic effective porosity is sufficient for practical use.

(2) The modified governing equation taking into account the vertical flow in the saturated

zone and dynamic effective porosity can accurately predict experimental dispersion relations,
highlighting the importance of the dynamic effective porosity in modeling watertable
fluctuations. For a given soil, the dynamic effective porosity decreases with increasing
fluctuation frequency, leading to a decrease in vertical water exchange between the saturated
zone and unsaturated zone and hence watertable waves can propagate further landward,
especially for high frequency fluctuations.

(3) In addition to watertable fluctuations, the dynamic effective porosity further plays an

important role in controlling seawater intrusion. As confirmed by laboratory experimental
data, by including the dynamic effective porosity, the numerical model predicts a more
landward seawater-freshwater interface and a higher position of the upper saline plume. This
suggests that neglecting the dynamic effective porosity leads to inappropriate estimations of
seawater intrusion in coastal unconfined aquifers.

The empirical expression for the dynamic effective porosity proposed here is based on

numerical results, and data from 1D sand column experiments. More measured data, obtained
under a variety of conditions, would be helpful to further examine the expression for the
dynamic effective porosity. For example, the dynamic effective porosity could be related to
pore water velocity given that the boundary signal is usually irregular (combination of tides
and waves), and the watertable fluctuation amplitude. Although uncertainties due to these



factors could provide further insights, despite its empirical basis this study shows that the
dynamic effective porosity plays an important role in modeling watertable fluctuations and
seawater intrusion. The real-valued effective porosity model presented here leads to reliable
predictions of groundwater response, which is essential for understanding many groundwater-
dependent processes in coastal unconfined aquifers.



**Appendix**

Assuming a single-component sinusoidal tide at the sea boundary, we have,

$$h(0,t) = D + A\sin(\omega t) \tag{A1}$$

Based on the dispersion relation, the analytical solution of $h$ is (Parlange & Brutsaert, 1987),

$$h = D + A\exp(-k_r Dx)\sin(\omega t - k_i Dx) \tag{A2}$$



**Code/Data availability**

The paper is theoretical, and data used in this study can be found in Parlange et al.

(1984), Cartwright et al. (2003, 2005), Shoushtari et al. (2016), Pozdniakov et al. (2019) and

Shen et al. (2020).



**Author contribution**

All authors contributed to the design of the research. ZL carried out data collation,

developed the theories and prepared the manuscript with contributions from all co-authors.

All authors contributed to the interpretation of the results and provided feedback.



**Competing interests**

The authors declare that they have no conflict of interest.





**Acknowledgments**
This research was supported by the National Natural Science Foundation of China
(51979095) and Marine Science and Technology Innovation Project of Jiangsu
(JSZRHYKJ202105). ZL acknowledges EPFL for financial support and JK acknowledges
the Qing Lan Project of Jiangsu Province (2020). We gratefully appreciate N. Cartwright for
providing the experimental data of Shoushtari et al. (2016), P. Wang for providing the
numerical data of Pozdniakov et al. (2019), and Y. Shen for providing the experimental data
of Shen et al. (2020).





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



**Table 1.** Soil properties for watertable fluctuation and seawater intrusion experiments.

|  | $K_s$ (m/s) | $n_e$ | $\alpha^{\,c}$ (m$^{-1}$) | $n^{c}$ | $\alpha_1$ (m$^{-1}$) | $n_1$ | $H_{\psi}$ (m) |
|---|---|---|---|---|---|---|---|
| Watertable fluctuation[a] | $4.7 \times 10^{-4}$ | 0.32 | 1.7 | 9 | 1.63 | 8.27 | 0.61 |
| Seawater intrusion[b] | $3 \times 10^{-3}$ | 0.4 | 11 | 6 | 10 | 5.23 | 0.1 |

[a]Compiled from Shoushtari et al. (2016)

[b]Compiled from Shen et al. (2020)

[c]$\alpha$ and $n$ are the VG fitting parameters (van Genuchten, 1980)



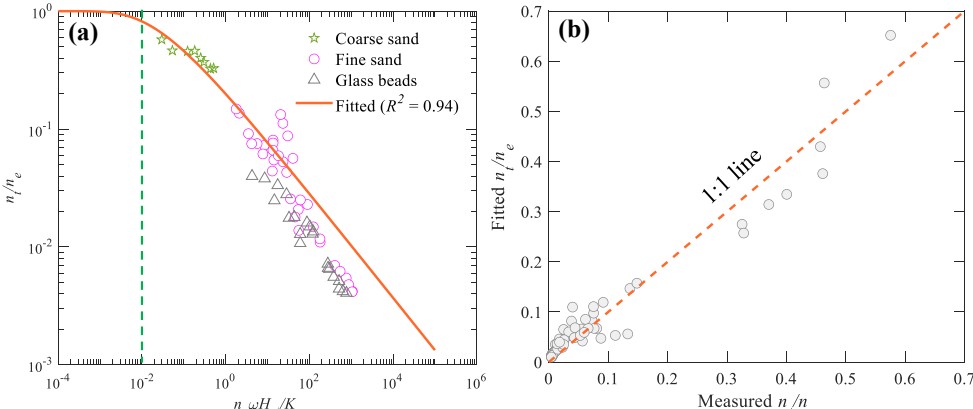

**Figure 1.** (a) Comparison of experimental and fitted relations between $n_t/n_e$ and $\tau_\omega$. Note that

the green dashed line indicates the critical value (about 0.01) when the effective porosity will

be significantly impacted by watertable fluctuations. (b) Fitted $n_t/n_e$ versus experimental $n_t/n_e$.

Experimental data are compiled from Cartwright et al. (2005).

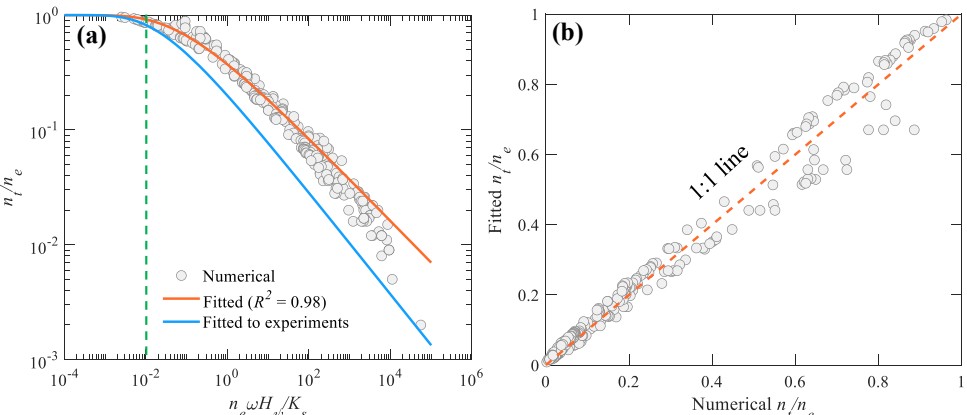


**Figure 2.** (a) Comparison of numerical and fitted relations between $n_t/n_e$ and $\tau_\omega$. Note that

the green dashed line indicates the critical value (about 0.01) when the effective porosity will
be significantly impacted by watertable fluctuations and the blue line is fitted to experimental
data of Cartwright et al. (2005). (b) Fitted $n_t/n_e$ versus numerical $n_t/n_e$. Numerical data are
compiled from Pozdniakov et al. (2019).



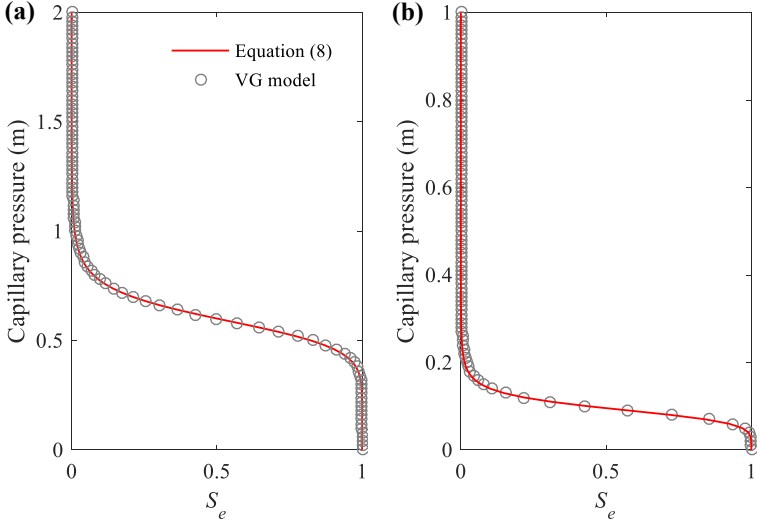

**Figure 3.** Comparison between equation (8) and the VG model for the soil adopted in (a)

watertable fluctuation and (b) seawater intrusion experiments. Data are compiled from

Shoushtari et al. (2016) and Shen et al. (2020), respectively



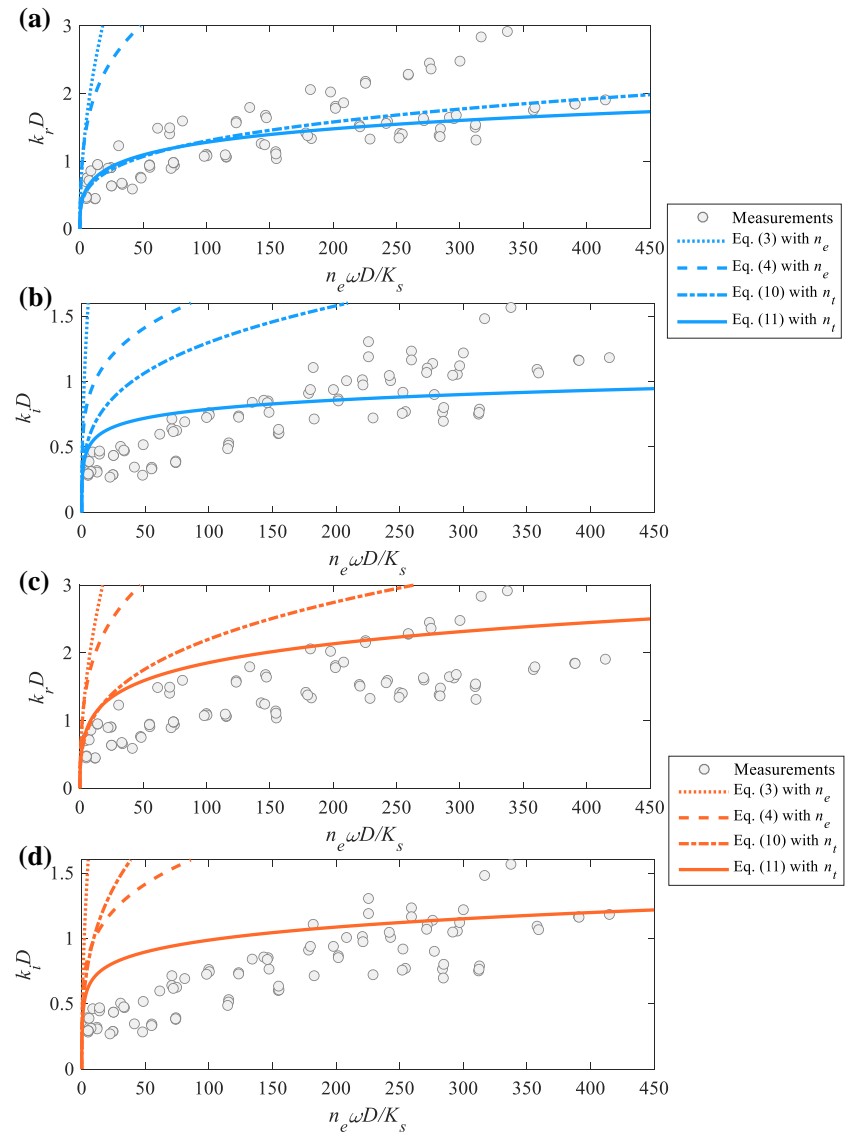

**Figure 4.** Comparison model predictions and experimental results of (a, c) amplitude decay

rate ($k_rD$) and (b, d) phase lag increase rate ($k_iD$) versus $n_e\omega D/K_s$. The values of $a$ and $b$ used

for the blue lines and orange lines are based on the experiments ($a = 0.0335$, $b = 0.4444$) and

numerical simulations ($a = 0.1216$, $b = 0.3642$), respectively. Parameters used are consistent

with Shoushtari et al. (2016): $D = 0.92$ m, $H_\psi = 0.61$ m and $K_s = 4.7 \times 10^{-4}$ m/s.



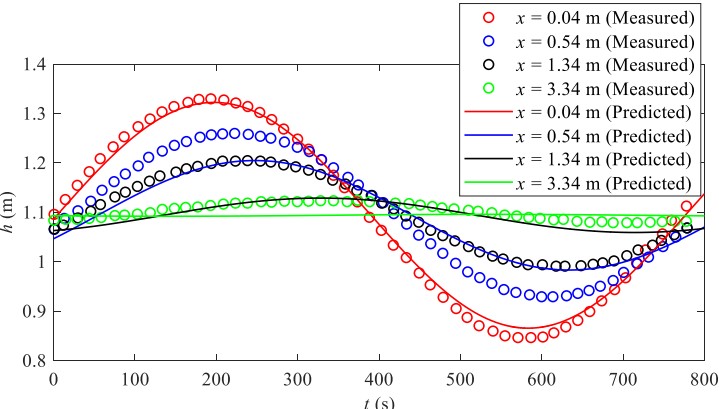


**Figure 5.** Comparison of measured watertable and predictions from equation (11).

Experimental data are compiled from Parlange et al. (1984). Since $n_e$ and $K_s$ were not
measured in the experiments of Parlange et al. (1984), the optimal value of $n_t/K_s$ in equation
(11) was estimated to be 0.5 s/cm based on the measured watertable at different locations.





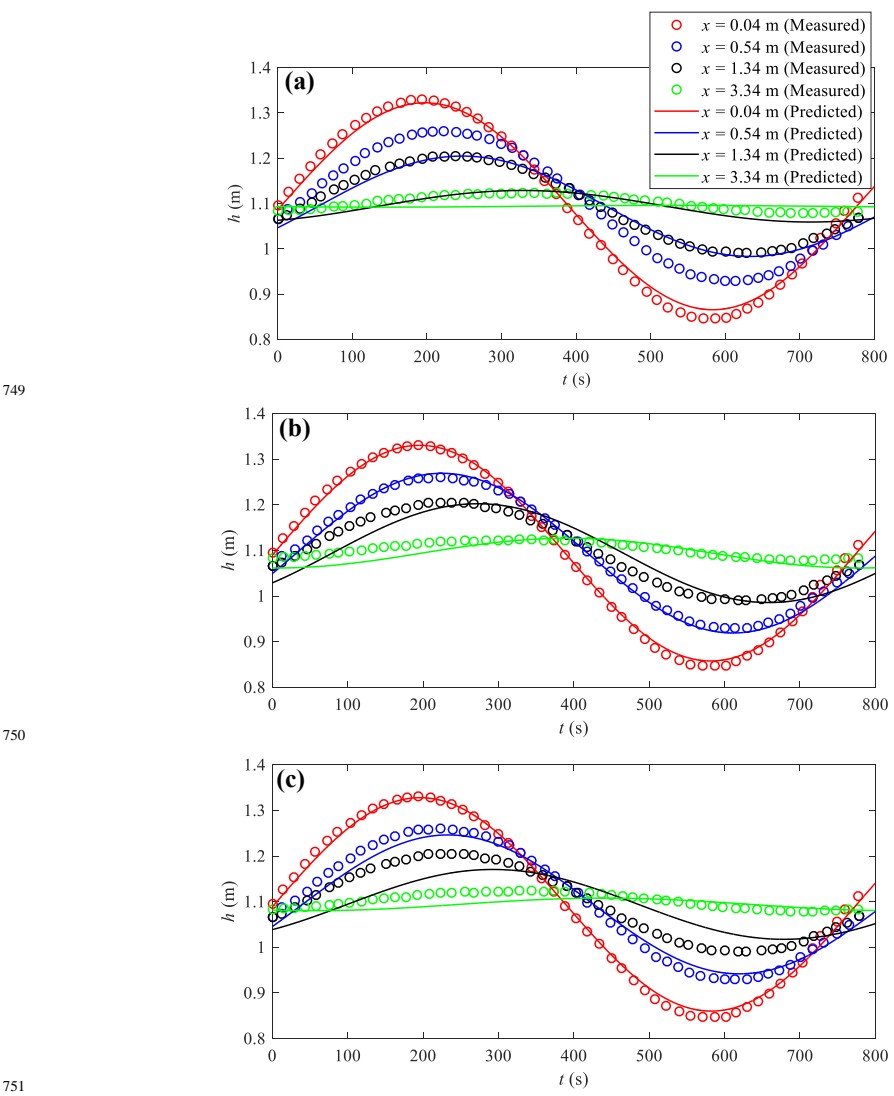




**Figure 6.** Comparison of measured watertable and predictions from (a) equation (2) (ignoring

the dynamic effective porosity), (b) equation (11) with $a$ = 0.0335 and $b$ = 0.4444, and (c)

equation (11) with $a$ = 0.1216 and $b$ = 0.3642. Experimental data are compiled from

Cartwright et al. (2003).

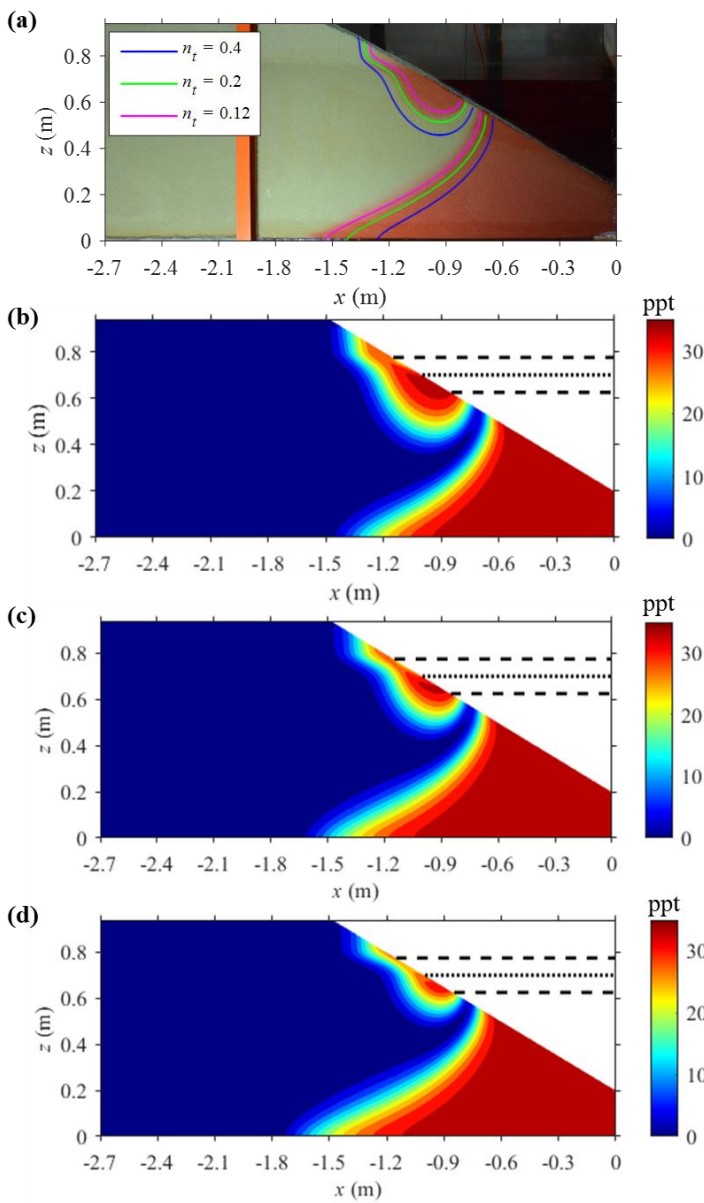

**Figure 7.** Comparison of (a) measurements (Shen et al., 2020) and numerical results (quasi-

steady state) with (b) $n_t = 0.4$ (equal to $n_e$), (c) $n_t = 0.2$ and (d) $n_t = 0.12$. Three lines in plot (a)

are simulated 50% isohalines corresponding to three cases respectively. In plots (b)-(c), the



upper and bottom dashed lines indicate the highest and lowest water levels, respectively, and

the middle dotted line represents the mean water level.