# Peer review of "Supporting Information for"

_Hydrology and Earth System Sciences, 2021_

## Author Comment (AC1)

[Figure]

**Figure 5.** Comparison of measured watertable and predictions from equation (11).

Experimental data are compiled from Parlange et al. (1984). Since $n_e$ and $K_s$ were not

measured in the experiments of Parlange et al. (1984), the optimal value of $n_t/K_s$ in equation

(11) was estimated to be 0.5 s/cm based on the measured watertable at different locations.

---

## Author Comment (AC2)

Responses to Referee #1's comments

Firstly, I note that this is the third time I am reviewing this work which has been previously submitted to and rejected from two other journal publications. My key concerns remain (see detailed comments below) and consequently I am unable to recommend publication of the submitted paper.

Thanks to the public review process of *HESS*, we have the opportunity to clarify the history of this work.

Versions of this work were rejected by two other journals (*WRR* and *JH*). However, contrary to the implication of reviewer's text, the manuscript has changed. Furthermore, for both journals, the other referees were positive concerning the manuscript.

This work solves a problem that was previously unexplained – this is the contribution of this work. The reviewer does not acknowledge this unfortunately. This is a fundamental point that should be considered and the reviewer's points do nothing to change this.

Abstract

li 27: "After validation with 1D experimental data and numerical simulations ..."

We will change to "After comparing with both experimental data and numerical simulations".

li 31-33: "accounting for vertical flow" - the equations used are the 2nd order approximation correction for vertical flows, there is an infinite order expression that has been excluded from the analysis (cf Nielsen et al, 1997). It is anticipated that the later (more accurate) correction vertical flow effects will not yield such favorable results.

There is no point to consider the infinite-order expression of Nielsen et al. (1997) given that its predictions do not agree with experimental data. For example, Shoushtari et al. (2016), who considered the infinite-order expression, state that "in the high frequency limit ($n\omega d/K \rightarrow 1$): (1) the theory predicts a zero phase lag ($k_i = 0$) corresponding to a standing wave scenario and (2) the amplitude decay rate has an asymptotic value of $k_r = \pi/2d$. The present data however will be shown to contradict this with both $k_r$ and $k_i$ observed to monotonically increase with increasing $n\omega d/K$". The modified effective porosity presented in the manuscript does nothing to change this finding. Furthermore, as shown by Shoushtari et al. (2016) in their Fig. 4, the **only** theory that agrees with experimental data is the so-called second-order theory. Thus, we considered only this latter theory in this manuscript.

li 35-38: "the phase lag can be ignored" The observations in Table 6 of Shoushtari et al (2017)

indicates that, for a 2DV propagating groundwater wave system, the phase lag in moisture content fluctuations is much greater than that for the watertable fluctuations. In addition, the dynamic effective porosity presented by the authors (eq 7) is derived based on a 1D sand column system so this is somewhat contradictory. I note that a reviewer in a previous submission was also critical of this assumption.

There is no contradiction. We are **not** predicting soil moisture in the vadose zone, which the reviewer is referring to. Indeed, this manuscript does not concern this, rather watertable fluctuations and seawater intrusion. As stated in the manuscript: "Results show that the Boussinesq equation accounting for the vertical flow in the saturated zone and dynamic effective porosity can accurately predict experimental dispersion relations (that all existing theories fail to predict), highlighting the importance of the dynamic effective porosity in modeling watertable fluctuations in coastal unconfined aquifers. This in turn confirms the utility of the real-valued expression of the dynamic effective porosity. An outcome is that the phase lag between the total moisture (above the watertable) and watertable height measured in laboratory experiments using vertical soil columns (1D systems) can be ignored when predicting watertable fluctuations in coastal unconfined aquifers (2D systems)." We see no reason to change this text.

li38-41: this has long been known.

Please give citations that prove the dynamic effective porosity facilitates watertable waves propagating further landward.

Highlights

1. the "modified expression" is the same as Pozdniakov et al. (2019). Equation 7a is the same as Pozdniakov et al. (2019), it has just been written using different notation. If you insert the authors' equations 9 and 7b into 7a you get the same equation as when you insert Pozdniakov et al's eq 12 into eq 15. The author's fitting parameter a being equivalent to 2.pi.f(l,m)/tau0 in the notation of Pozdniakov et al. (2019). Therefore the correct description of what has been done is "Here we introduce the existing formulation of Pozdniakov et al. (2019) using different notation ..."

We note that the reviewer recognizes that this expression was derived in a different way later.

This is a semantic point essentially. In Pozdniakov et al. (2019), $f(l,m)$ is an approximate expression (their equation 12) whereas in this manuscript it is an exact expression derived using the modified van Genuchten model (equation 8). So, it is incorrect to assert that it is a matter of "different notation". It is, on the other hand, accurate to describe our formula as a modification of their formula, as done in the text.

2. Only for the 2nd order approximation, the infinite-order correction for vertical flow effects has been omitted by the authors

Please see our reply at the bottom of Pg. 1.

Main Body

li 118-119: review language. It is known and agreed upon that the unsaturated zone does affect water table fluctuations. The dynamic effective porosity is a way of parameterizing this affect. Therefore it follows that the dynamic effective porosity will affect water table fluctuations. It is the extent to which, and our ability to correctly quantify the dynamic effective porosity that remains unclear.

We will correct this sentence.

li 120-121: I agree that using a complex effective porosity in a practical application (e.g. numerical model) is not possible but note that Cartwright et al (2006) overcame this by using the absolute value of the complex number which led to reasonable outcomes for practical use.

We will mention this work in Introduction. Cartwright et al (2006) adopted an absolute value of the complex number and got reasonable outcomes. Their approach further confirms that the phase lag between the total moisture (above the watertable) and watertable height can be ignored when predicting watertable fluctuations in coastal unconfined aquifers.

li 122-124: The influence of water table fluctuations on salt water intrusion is significant at long time scales (e.g. tidal - Robinson et al, storm surges - Cartwright et al). At these longer time scales, the influence of the unsaturated zone (and hence the dynamic effective porosity) on water table fluctuations becomes negligible (refer to all available dispersion relation theory and even the authors results showing that their nt/ne ~ 1 for small values of Tau_w).

We disagree. Kong et al. (2015) showed that unsaturated flow effects play an important role in affecting watertable fluctuations with long time scales.

li 153-1161: The authors acknowledge that this is a 2nd order correction for vertical flow effects, however Nielsen et al (1997) also provide an infinite-order solution which should be included in the analysis. I note that it is included in a supplementary figure S2 but it should be added to Figure 4 with the authors nt expression in place of ne. I anticipated that this will yield a much poorer comparison with the data and will highlight the somewhat fortuitous outcome that the 2nd order solution provides a reasonable comparison.

Please see our reply at the bottom of Pg. 1.

li 181-193: It is not clear to me how the equation is modified from the existing. Whilst the authors' modified dynamic effective porosity has been derived differently the result is the same as Pozdniakov et al. (2019) albeit with a different notation. If you insert the authors' equations 9 and 7b into 7a you get the same equation as when you insert Pozdniakov et al's eq 12 into eq 15. The authors' fitting parameter "a" being equivalent to 2.pi.f(l,m)/tau0 in the notation of Pozdniakov et al. (2019).

Please see our reply on Pg. 2 starting with "We note that the reviewer recognizes …"

li 199: are they wetting or drying curves?

We will clarify that they are drying curves.

li 207-209: I would argue that the authors' approach is also approximate because, ultimately at the end their eq 7 is semi-empirical and requires fitting to data.

The reviewer has apparently misread the text. The cited lines (again) point out the difference in derivations as described in our reply on Pg. 2 starting with "We note that the reviewer recognizes …".

li 255-257: In figure 1 there is a clear departure between the curve fit and the data as $n_w H\psi/K$ increases (and $n_t/n_e$ decreases) indicating poor performance where the influence of the unsaturated zone on water table fluctuations is large (ie small $n_t$).

There is a deviation, yes, but we consider it to be acceptable for the effective porosity in terms of results. Indeed, the Boussinesq equation and numerical model with the dynamic effective porosity calculated from Eq. 7a accurately predict watertable fluctuations (Figures 4-6) and seawater intrusion (Figure 7), respectively.

li 258-262: the limited ability of numerical solutions to Richards' equation to reproduce the lab data when neglecting hysteresis is discussed in depth in Cartwright et al (2005)

We will mention this work.

li 288: Nielsen and Perrochet (2000a,b) first proposed the complex effective porosity concept

We will correct this.

li 290-292: It is important to clarify that, regardless of whether the system is 1D or 2DV, water table fluctuations are induced by external forcing at a boundary (ocean tides, wave, atmospheric pressure ...). Moisture content may, or may not, play a role in influencing the nature and extent of the response. Also note that the phase lag between moisture content fluctuations in the unsaturated and those in the water table, is also present in 2D systems

(Shoushtari et al, 2017).

It is obvious that watertable fluctuations are "induced by external forcing at a boundary", as already indicated in the text. We see no point to repeat this at these lines.

Concerning the phase lag, see our reply at the top of Pg. 2.

li 298: cite the source of the experiments

We will add the references (i.e., Parlange et al., 1984; Cartwright et al., 2003; Shoushtari et al., 2016).

li 342: as per my earlier comment, I anticpate that if the infinite-order solution was used rather than the 2nd order one the comparison will be much worse. Please add these curves to your Figure 4.

Please see our reply at the bottom of Pg. 1.

li 347-349: I disagree. As per my earlier comments, it is rather fortuitous that the 2nd order solution seems to do OK.

Please see our reply at the bottom of Pg. 1.

li 353-356, Fig 5 and 6: For a more rigorous comparison, rather than time series, present both the amplitude and phase profiles (ie A vs x and phase lag vs x)

Since we are predicting watertable fluctuations, it is more direct to present time series of watertable in different locations.

sec 3.3: It seems to me that the approach adopted to examine the influence of the dynamic effective porsoity (ie the link between saturated and unsaturated zones) on saltwater intrusion is fundamentally flawed. As described, SUTRA solves the variably-saturated flow equations so therefore the moisture exchange between saturated and unsaturated zones is implicitly accounted for in the governing equations. To then replace the storage coefficient with a reduced dynamic effective porosity does not make sense phsyically as you are essentially accounting for the exchange twice.

The static effective porosity (equal to the saturated soil water content) is also used in Richards' equation. Therefore, the static effective porosity also can be corrected by replacing the dynamic effective porosity. A recent *WRR* paper (Zheng et al., 2022) confirms that this approach is reasonable. It should in particular be noted that our dynamic effective porosity was adopted in this paper and the authors concluded "Simulation using both Cartwright's 'wetting and drying' model and *Richards' model with dynamic effective porosity* are used to

evaluate experimental results, with the latter model providing a better match for large capillary-fringe truncation".

**References**

Cartwright, N., O. Z. Jessen and P. Nielsen (2006). "Application of a coupled groundsurface water flow model to simulate periodic groundwater flow influenced by a sloping boundary, capillarity and vertical flows." Environmental Modelling & Software 21(6): 770-778.

Shoushtari, S. M. H. J., N. Cartwright, P. Perrochet and P. Nielsen (2017). "Two dimensional vertical moisture-pressure dynamics above groundwater waves: Sand flume experiments and modelling." Journal of Hydrology 544: 467-478.

Nielsen, P. and P. Perrochet (2000). "Watertable dynamics under capillary fringes: experiments and modelling." Advances in Water Resources 23(5): 503-515.

Nielsen, P. and P. Perrochet (2000). "ERRATA: Watertable dynamics under capillary fringes: experiments and modelling [Advances in Water Resources 23 (2000) 503-515]." Advances in Water Resources 23(8): 907-908.

Pozdniakov, S. P., et al. (2019). "An Approximate Model for Predicting the Specific Yield Under Periodic Water Table Oscillations." Water Resources Research 55(7): 6185-6197.

**References**

Cartwright, N., Jessen, O. Z., & Nielsen, P. (2006). Application of a coupled ground-surface water flow model to simulate periodic groundwater flow influenced by a sloping boundary, capillarity and vertical flows. Environmental Modelling & Software, 21(6), 770-778. https://doi.org/10.1016/j.envsoft.2005.02.005

Cartwright, N., Nielsen, P., & Dunn, S. (2003). Water table waves in an unconfined aquifer: Experiments and modeling. Water Resources Research, 39(12), 1330. https://doi.org/10.1029/2003WR002185

Kong, J., Xin, P., Hua, G.-F., Luo, Z.-Y., Shen, C.-J., Chen, D., & Li, L. (2015). Effects of vadose zone on groundwater table fluctuations in unconfined aquifers. Journal of Hydrology, 528, 397–407. https://doi.org/10.1016/j.jhydrol.2015.06.045

Pozdniakov, S. P., Wang, P., & Lekhov, V. A. (2019). An approximate model for predicting the specific yield under periodic water table oscillations. Water Resources Research, 55(7), 6185–6197. https://doi.org/10.1029/2019WR025053

Shoushtari, S. M. H. J., Cartwright, N., Nielsen, P., & Perrochet, P. (2016). The effects of oscillation period on groundwater wave dispersion in a sandy unconfined aquifer: Sand flume experiments and modelling. Journal of Hydrology, 533, 412–420. https://doi.org/10.1016/j.jhydrol.2015.12.032

Zheng, Y., Yang, M., & Liu, H. (2022). The effects of truncating the capillary fringe on water-table dynamics during periodic forcing. Water Resources Research, 58(1), e2021WR031112. https://doi.org/10.1029/2021WR031112

---

## Author Comment (AC3)

**Responses to Referee #2's comments**

The study is keyed to proposing an empirical expression to evaluate a dynamic effective porosity and assess its impact on the quantification of watertable fluctuations and seawater intrusion in coastal aquifers. After studying the work, I am afraid I am not in a position to recommend publication at this stage. In addition to having some doubts about the possibility that this study constitutes more than an incremental advancement, at least the way it is framed and the way the Authors present it, I do have two major concerns. When combined, these seriously question the validity of the approach and of the key results of this work.

We use the concept of a dynamic effective porosity to resolve a problem highlighted in the recent paper of Shoushtari et al. (2016). As further explained below, these authors presented experimental data on coastal aquifer watertable fluctuations that could not be modelled by existing analytical results or by Richards' equation. The dynamic effective porosity is shown to produce results that agree reasonably with a range of data from careful experiments.

The Authors observe that considering vertical flow effects making use of (i) an approximated (at second-order) formulation and (ii) a dynamic effective porosity leads to an accurate prediction of experimental dispersion relations of watertable waves. This result is in contrast with a previous analysis according to which it is shown that an infinite-order expression (that includes the second-order approximation presented in this study) cannot predict these results in an accurate way. In order to resolve this issue the authors should compare their results as well as the infinite-order expression against outcomes of the Richards' equation (which accounts for vertical flow under variably saturated flow settings). It can also be noted that, in addition to the theoretical elements described above, the physical basis according to which an approximate solution should provide improved results as opposed to its exact counterpart is not clear.

As we explain below, the reviewer asks that we repeat work that is already well documented in the literature (Shoushtari et al., 2016), and is described in the manuscript (lines 104-111, 314-319).

The reviewer mentions approximations presented by Nielsen et al. (1997). These authors considered fluctuations in an aquifer with a vertical boundary at which periodic (hydrostatic) fluctuations are imposed. Effects of the vadose zone above the watertable were not considered. The dispersion relations referred to by the reviewer were designated by Nielsen et al. (1997) as the "second-order small amplitude" and "infinite-order small amplitude" equations (Nielsen et al., 1997, their Eqs. 16 and 17, respectively). Nielsen et al.'s approximations, along with other similar approximations (i.e., shallow, capillarity free aquifer, shallow with capillarity effects, non-shallow, capillarity free, non-shallow with capillarity effects), were already compared with experimental data (Shoushtari et al., 2016, their Fig. 4), and with numerical solutions of Richards' equation (Shoushtari et al., 2016, their Fig. 8), viz.,

- Experimental data. Referring to their Fig. 4, Shoushtari et al. (2016) state "The data indicates a monotonic increase in wave number with increasing $n\omega d/K$ which is in direct contrast with most of the dispersion relations which predict: (1) zero phase lag ($k_i = 0$)." Indeed, the only theory that predicts the non-zero phase lag of the experimental data is the second-order theory. Shoushtari et al. (2016) described the second-order theory as "The observed monotonic increase in wave numbers with increasing $n\omega d/K$ appears to be captured by the 2nd order (in $n\omega d/K$), capillarity free dispersion relation." The main difference is that, in contrast to the second-order approximation, the other theories (including the infinite-order theory of Nielsen et al., 1997) predict standing wave behavior for large $n\omega d/K$. We emphasize that the second-order theory is the only theory that gives predictions that agree with the experimental data, i.e., the experimental data of Shoushtari et al. (2016) serve to validate the second-order theory.

- Numerical solutions to Richards' equation. This was also done by Shoushtari et al. (2016, their Fig. 8), who presented results with and without hysteresis included. Like the infinite-order approximation, the Richards' equation results did not agree with the experimental data: "Whilst the inclusion of hysteresis effects have led to some small quantitative differences when compared to the non-hysteretic results, both sets of results show the same qualitative discrepancy when compared to the data" (Shoushtari et al., 2016). This means that the standard Richards' equation model should not be used to simulate propagation of high-frequency fluctuations.

The reviewer's final sentence refers to the infinite-order approximation as the "exact counterpart" of the second-order approximation. However, the infinite-order approximation is not exact if one considers that it is based on the Boussinesq approximation, assumes small amplitude fluctuations, ignores the vadose zone, etc. As indicated in the present study, the vertical water exchange between the saturated zone and unsaturated zone varies with the fluctuation frequency. Repeated analytical corrections without accounting for unsaturated zone effects may lead to an inappropriate estimate to the hydrostatic pressure and hence deviations between predicted and experimental results.

I found the approach adopted in modeling the saltwater intrusion not convincingly supported. To the extent of my knowledge, the code adopted (SUTRA) already solves variable saturated (saturated-unsaturated ) flow settings. Therefore, while a model parameter such as a dynamic effective porosity could be considered and included to account for the effects of the unsaturated zone on water table dynamics when these effects have not yet been considered (e.g., when using saturated models such as Eq.1-2 of the manuscript), I strongly doubt about its use and physical implications when solving the Richards' equation. The latter already accounts for the unsaturated zone and its impact on subsurface flow dynamics. As such, I find the approach to be inconsistent and not substantiated by robust physical bases.

Richards' equation models unsaturated flow, however, it fails to describe unsaturated flow in some situations. For example, as indicated by Cartwright. (2014), when predicting the water retention curve above watertable fluctuations, Richards' equation matches well with

experimental results for low-frequency watertable fluctuations, but it deviates markedly from the measurements for high-frequency watertable fluctuations (Fig. R1). This means that Richards' equation cannot accurately predict the water exchange between the saturated zone and unsaturated zone for high-frequency watertable fluctuations, as already reported by Cartwright. (2014) and Shoushtari et al. (2016, 2017). Specifically, as the fluctuation period decreases from 12.25 h to 75 s, the maximum soil water content measured at $z = 1.2$ m decreases from the saturated soil water content (0.355) to a smaller value (about 0.22), but the predicted values nearly keep the same (i.e., equal to the saturated soil water content). In this manuscript, we rectified this shortcoming of Richards's equation by introducing the dynamic effective porosity. The comparison between measured and predicted results shows that this method works well (Fig. 7). In addition, Zheng et al. (2022) confirmed the validity of this approach. In their paper, Zheng et al. (2022) used our dynamic effective porosity expression and found that "Simulation using both Cartwright's 'wetting and drying' model and *Richards' model with dynamic effective porosity* are used to evaluate experimental results, with the latter model providing a better match for large capillary-fringe truncation". In summary, the "standard" Richards' equation has already been shown not to simulate observed vadose zone pressure head-saturation data. On the other hand, inclusion of the dynamic effective porosity in Richards' equation produces results that do agree with experimental data.

[Figure]

**Fig. 2.** Observed moisture-pressure relationships (bold solid lines) for a full oscillation period at $z = 1.2$ m and 1.4 m. Each panel corresponds to a different oscillation period as indicated. The dashed lines denote the equilibrium van Genuchten (1980) curves (drying and wetting) based on parameters given in Table 1. The dash–dot curve is an indicative van Genuchten (1980) curve with $\beta = 3$.

[Figure]

**Fig. 3.** Simulated moisture–pressure relationships (bold solid lines) at $z = 1.2$ m and 1.4 m. Each panel corresponds to a different oscillation period as indicated. The dashed lines denote the equilibrium van Genuchten (1980) curves (drying and wetting) based on parameters given in Table 1. The dash–dot curve is an indicative van Genuchten (1980) curve with $\beta = 3$. Simulations used the hydraulic parameters given in Table 1 and bottom forcing parameters of $A_0 = 0.15$ m and $d = 0.9$ m.

**Fig. R1.** Comparison of measured and predicted water retention curves (Cartwright, 2014, Figs. 2, 3). Richards' equation matches well with experimental results for low-frequency watertable fluctuations, but it deviates markedly from experimental results for high-frequency watertable fluctuations.

**References**

Cartwright, N. (2014). Moisture-pressure dynamics above an oscillating water table. Journal of Hydrology, 512, 442–446. https://doi.org/10.1016/j.jhydrol.2014.03.024

Nielsen, P., Aseervatham, R., Fenton, J. D., & Perrochet, P. (1997). Groundwater waves in aquifers of intermediate depths. Advances in Water Resources, 20(1), 37–43. https://doi.org/10.1016/S0309-1708(96)00015-2

Shoushtari, S. M. H. J., Cartwright, N., Nielsen, P., & Perrochet, P. (2016). The effects of oscillation period on groundwater wave dispersion in a sandy unconfined aquifer: Sand flume experiments and modelling. Journal of Hydrology, 533, 412–420. https://doi.org/10.1016/j.jhydrol.2015.12.032

Shoushtari, S. M. H. J., Cartwright, N., Perrochet, P., & Nielsen, P. (2017). Two-dimensional vertical moisture-pressure dynamics above groundwater waves: Sand flume experiments and modelling. Journal of Hydrology, 544, 467–478. https://doi.org/10.1016/j.jhydrol.2016.11.060

Zheng, Y., Yang, M., & Liu, H. (2022). The effects of truncating the capillary fringe on water-table dynamics during periodic forcing. Water Resources Research, 58(1), e2021WR031112. https://doi.org/10.1029/2021WR031112